# Construction of CPW Pogo Pin Probes for RFIC Measurements

**DOI:** 10.3390/s25061677

**Published:** 2025-03-08

**Authors:** K. M. Lee, J. S. Kim, S. Ahn, E. Park, J. Myeong, M. Kim

**Affiliations:** 1School of Electrical Engineering, Korea University, Seoul 02841, Republic of Korea; leemin7@korea.ac.kr (K.M.L.);; 2Samsung Electronics Co., Ltd., Suwon 16677, Republic of Korea; sanguck.ahn@samsung.com (S.A.);

**Keywords:** RF pogo pin probe, tilted pogo pin structure, 250 μm linear aligned pitch, custom calibration standard for pogo pin, pogo pin modeling for calibration

## Abstract

A new radio frequency (RF) probe using pogo pin tips for integrated chip (IC) measurement up to 50 GHz is proposed. It offers high durability due to the pogo pins and meets three key design criteria for general IC measurement: (1) a 45° tilted shape with a 70 μm tip protrusion for easy microscope inspection, (2) linear pogo pin alignment for commercial chip pad contact, and (3) a 250 μm pitch compatible with standard IC pad pitches. This design is distinct from traditional pogo pin probe cards which place pogo pins in vertical form, in a diagonal arrangement, and at wide intervals. The probe exhibits a low insertion loss of 1.6 dB at 45 GHz. A printed circuit board (PCB)-based calibration standard for the calibration of the designed probe is constructed, which is adjusted to inductance and capacitance values using a simulation to form the Vector Network Analyzer (VNA) calibration set. The measurements of a commercial amplifier IC using this probe show a nearly identical performance to commercial RF probes, confirming its accuracy and reliability.

## 1. Introduction

In general, an RF probe including a micro-sized needle tip is used to measure the integrated chips above GHz, and it is measured by contacting the pad of the IC. However, these common micro-sized needle tips in RF probes have a high risk of shattering by overdrive [1,2]. Therefore, the probes are generally not recommended for measuring nonplanar surfaced circuits. A formfactor air coplanar probe (ACP) that can attempt nonplanar circuit measurement has been developed in advance, but the overdrive distance is still limited due to remaining shattering risk [3,4].

As the demand for durable probe tips has increased, studies have been conducted on pogo pins using spring structures that are more durable and cost effective than conventional micro-sized needle tips. While pogo pins were primarily used in DC or MHz bands, pogo pins have been shown to operate up to tens of GHz [5], with studies confirming signal transmission characteristics up to 40 GHz [6,7]. However, these studies are limited to wide pitch configurations, leaving uncertainties at narrower pitches. Recent efforts have investigated pogo pin arrays in sockets supporting customized ball maps to assess their performance at narrow pitches.

Microwave studio simulations have shown that socket-loaded pogo pin arrays operate up to 42 GHz [8]. Optimized socket designs were developed to reduce reflection loss and crosstalk in both single and double pin arrangements, with measurements up to 10 GHz [9,10]. Other studies demonstrated signal transmission up to 20 GHz [11]. As the pogo pin pitch narrows, socket structures with additional ground pins, guard bodies, and space transformers have been developed to minimize signal leakage [12,13,14]. The ground–signal–ground (G-S-G) pogo pin probe prototype is also reported to operate with a loss of −13 dB at 20 GHz [15]. Recently pogo pin probe cards with low-loss performance up to a maximum of 50 GHz, including millimeter-wave bands, are reported [16,17,18].

However, typical probe cards have two problems using general IC measurements. The first issue is the mounted pogo pin angle. Conventional pogo pin probes load pogo pins vertically because they are generally designed for use in automated measurement systems with additional alignment systems. Vertically aligned pogo pins are not visually identifiable, and alignment errors may occur due to the position offset of the vertically loaded pins [19,20,21]. The manual alignment of the probe tips in the correct position requires the visual exposure of the tips in the structure that holds the pins. The second issue is the pitch size between pins. Previous pogo pin structures used a minimum pin pitch of 350 μm since narrowing the pitch led to an increased risk of signal leakage [16,17]. However, the G-S-G probe configuration suitable for typical IC pads required a narrower pitch of 100 to 250 μm. Also, the typical diagonal ground pogo pin arrangement complicates pad contact. Therefore, a linear-aligned ground pin arrangement is essential. The commercial pogo pin probe satisfies the linear alignment but its wide pitch (e.g., 800 μm) limits its applicability for universal IC testing [22].

In order to use the probe for IC measurement, it is necessary to set the circuit reference condition to meet the probe tip contact condition with the calibration standard. In the case of RF probes, SOLT calibration is commonly used to meet short, open, load, and through conditions using a very sophisticated calibration standard. The general calibration standard is fabricated for micro-sized needle tip probes, resulting in low loss and no step difference to suit probe contacts. However, to provide these characteristics, the metal pattern of the calibration standard is made of gold [23]. There is a problem that metal patterns are easily worn because the probe is generally overdriven for accurate contact between the probe and the metal pattern and the durability of gold is low.

This paper proposes a new form of pogo pin probe that solves two problems of typical pogo pin probes. The proposed pogo pin probe structure, shown in Figure 1, can be used to measure commercialized RF chips by arranging pogo pins at an angle of 45° and arranging signal pins and ground pins in a linear arrangement. The designed probe has a reduced 250 μm pitch available to measure commercial ICs. A cost-effective calibration standard using a PCB to calibrate the designed pogo pin probe is also designed. The designed calibration standard is built for SOLT calibration. Since there is no problem even if there is a step due to the high durability of pogo pins, a calibration standard using chip resistance rather than film-type resistance is designed. The parameters for the VNA calibration set are extracted by analyzing the fabricated calibration standard. Finally, the commercial IC circuit is measured to verify the performance of the proposed probe.

## 2. Pogo Pin Probe Design

Figure 1 shows a photograph of the designed pogo pin probe and components of four parts of the pogo pin probe. Each component is a metal socket with a pogo pin array, a PCB for transition between the connector and pogo pin, an OS-50 connector, and a metal mounting frame for the positioner combination. The overall probe structure resembles the conventional RF probe structure. Compared with the typical probe, the probe tip part is replaced with a pogo pin socket. Pogo pins in contact with ICs are placed in the G-S-G array, with selected intervals of 250 μm for commercial chip testing. The pogo pin socket consists of a metal block part that makes up the coaxial line and an insulator that holds the pogo pin array. The metal block and the insulator are made of brass and polyamide-imide (PAI, dielectric constant 3.9) material, respectively, which are suitable for precision machining. Strong assemblies with the metal block and insulators are made by inserting four metal post structures inside the insulators.

The metal frame part for the positioner assembly is fabricated by aluminum. It is manufactured in a standard size to fit the commercial probe positioner. A guide pin structure is applied so that the main components can be placed at the correct location. The connector used an OS-50 product to transmit signals with low insertion loss up to the 50 GHz band. The PCB converter for signal transmission used a RF-35 substrate (dielectric constant 3.5). The thickness of the substrate is selected as 10 mil to withstand the pressure of the spring in the pogo pin. The PCB line is in direct contact with the connector pin to deliver the transmitted RF signal to the pogo pin, and a via wall is configured around the line for ground connection and minimum signal leakage in the high frequency band.

The detailed structure of the designed pogo pin and metal socket is shown in Figure 2. The length of the main metal block in the signal propagation direction is 1.3 mm, and the dielectric insulator is 0.7 mm. The difference from the typical pogo pin socket is that the pogo pin in this design is not mounted vertically but is mounted at an angle. The proposed pogo pin socket is designed at a 45° angle and is designed to enable accurate probing with a microscope when measured, like a normal RF probe. The pogo pin tip length of 250 μm is protruded, as shown in Figure 2, and can be visualized with a horizontal length of 70 μm or more, which is sufficient for manual IC measurement. At angles smaller than 45°, the length in the horizontal direction is short, making it difficult to identify pogo pin protrusions. This is because structures such as insulators obscure part of the pogo pin length. At angles above 45°, more protrudes in the horizontal length, but because the laid pogo pin receives a force vertically, the force in the pin compression direction is reduced, increasing the possibility that the pogo pin itself is damaged by a force during landing. Therefore, the angle of the pogo pin is selected at 45° to secure a 70 μm length that can be used to check the pin protrusion using a microscope and to facilitate pogo pin compression.

The pogo pin socket is comprised of a metal block and two insulator structures, as seen in Figure 2. The internal structure of the metal block is constructed in the form of a coaxial line. To form a 50 Ω coaxial line impedance, the diameter of the signal pin barrel and the metal block hole are selected to be 0.15 mm and 0.34 mm, respectively. The metal block also serves as a shielding structure for the pogo pin ensuring low-loss signal transmission. The diameter of the ground pin and metal block hole is the same as to provide fine ground connection. The pogo pin tips are placed in a linear G-S-G arrangement of 250 μm, shown in Figure 2, to support the pad pitch of a typical RF chip. To achieve both the 50 Ω metal block hole and 250 μm pin pitch, the required metal block thickness between the G-S-G holes is only 5 μm. Since it is impossible to manufacture such thin walls, a deformed hole structure is designed, as shown in Figure 2. The new hole merged the G-S-G holes with removed wall structures while still maintaining the coaxial line mode and 48 Ω impedance, which is verified by High Frequency Structure Simulation 2024 R2 (HFSS).

The insulator for fixing the pogo pin position is designed in a trapezoidal structure, as shown in Figure 2. This shape blocks the collision between two probes in a two-port measurement setup and secures a distance from the metal chuck where the IC is placed. The insulator fixes the barrel of the pogo pin and leaves the plunger free at each side by differently fabricated hole diameters. The material dielectric constant is 3.9 with a guaranteed firmness for fixation. However, to satisfy the 50 Ω characteristic impedance path with a 0.15 mm diameter pogo pin barrel at a 250 μm pitch, a lower dielectric constant of 2.2 is required and the material with a dielectric constant of 2.2 has very soft properties problems. Due to its soft properties, the structure may be damaged during fabrication and precise fabrication is impossible, making it difficult to manufacture the structure according to the desired numerical values. In order to correct the impedance falling due to the 3.9 dielectric constant, the diameter of the air gap is designed to be 0.2 mm, as shown in Figure 2, and it is confirmed that the impedance can be adjusted to near 50 Ω. This design prevents internal reflection in the pogo pin by maintaining impedance. The unremoved reflection signal can also degrade calibration accuracy.

The spacing of the pogo pins is basically designed as a pitch capable of measuring commercial RF chips. The pitch is selected within the range of 100 to 250 μm, which is widely used in commercial ICs. In addition, if the pitch is widened, there is a problem of leakage in terms of the field that occurs between the internal pins of the dielectric insulator. In contrast, there is also a problem in fabricating a probe with a pogo pin array having a too narrow pitch size. When excluding the diameter of the barrel at the pitch interval of 250 μm, very narrow intervals of 100 μm are formed between the barrels. The narrow spacing increases the difficulty of fabricating the insulator wall. Thinner pogo pins could reduce the gap, but manufacturing challenges arise. Considering both the manufacturing feasibility and the leakage problem, a 250 μm pitch is finally chosen.

## 3. Insertion Loss Measurement of Pogo Pin Probe

Figure 3a is a photograph of contacting the through pattern with the pogo pin probe and commercial probe (ACP40-A-GSG-150, Formfactor, Inc., Livermore, CA, USA) to measure the insertion loss performance of the pogo pin probe. Commercial probe measurements are also performed under the same conditions for performance comparison. To check the characteristics of the probe itself, subminiature version A (SMA) calibration is performed. This calibration uses the 2.4 mm 85056D calibration kit to calculate the value up to the reference plane of the probe connector. The result of measuring the back-to-back through line after SMA calibration is shown in Figure 3b. Figure 3b shows that about a 1.6 dB insertion loss occurs at 45 GHz. This insertion loss of the pogo pin probe is higher than the 0.33 dB (at 45 GHz) insertion loss of the commercial probe but the pogo pin measurement result shows a linear characteristic that the insertion loss increases with increasing frequency. However, it shows the problem of a periodic ripple. This ripple shows that higher reflection factors than design properties are occurring in the pogo pin structure.

The ripple has a period of approximately 6.5 GHz. The half-wavelength length of this frequency is similar to the distance between the input and output RF connector pins in the back-to-back measurement structure. This suggests that the reflection originates from the SMA-to-PCB transition. Therefore, the ripple is likely due to a mismatch at the connection between the SMA connector and the PCB line. To verify the cause, a back-to-back line structure composed only of a SMA-to-PCB converter is fabricated and measured. The structure comprises a SMA connector, PCB converter, transmission line, PCB converter, and SMA connector in order. Measurements confirmed that the ripple occurred with a period matching the half-wavelength of the verification structure. Although the line connection pad impedance in contact with the vertically connected connector pin is designed to be 50 Ω, PCB manufacturing precision issues may have led to inaccuracies in the etching of the via pad and location of the via pad. As a result, fabrication inconsistencies increased reflection loss.

## 4. Pogo Pin Probe Calibration for Vector Network Analyzer

Figure 4a shows the calibration standard pattern for the pogo pin probe calibration. It is designed in low-loss TLY-5 PCB substrates (dielectric constant 2.2) providing low material loss properties up to 50 GHz. Short, open, load, and through patterns capable of SOLT calibration are designed. The calibration standard pattern resembles commercial products. Unlike the typical calibration standard, the load pattern is designed with a nonplanar surface by soldering chip resistance (green blocks in Figure 4a, Vishay’s CH02016-100RGF, Vishay, Malvern, PA, USA). Since the pogo pin probe can handle height differences, probing soldered surfaces is allowed. The characteristics of the soldering resistors are confirmed before use in this study. The variation in the resistance used is not large (50~50.3 Ω), and the load pattern closest to 50 Ω is used for actual verification. The through pattern is designed as a coplanar waveguide (CPW)-type line to support G-S-G pin contact, and it is designed to be 3.3 mm long to provide a 3 ps standard delay. The width of the CPW line is 0.282 mm, and the gap between the line and ground is 0.05 mm to support the 250 μm pin pitch. Since the probe tip is durable, direct pin contact can also be used as a through calibration, which can minimize the delay to near 0 ps. Short and open patterns with a short length of 0.3 mm are designed for both the minimized parasitic components and adequate probing area.

SOLT calibration involves establishing a reference point by inputting the ideal SOLT values into the VNA through probe contact. However, the fabricated calibration standard cannot achieve ideal values. In the short pattern, a series inductance component arises due to the distance between the signal and ground pins in the CPW structure. In the open pattern, capacitance forms between calibration patterns, and the load may deviate from the intended 50 Ω terminal impedance. Additionally, achieving zero delay in through measurements is difficult. To address these limitations, two simulations are conducted to construct an accurate calibration set of the fabricated calibration standard.

The first simulation is contacting SOLT patterns, including pogo pin probe tips and probe pads, using HFSS. These simulation results include the capacitance, inductance, and delay of the actual SOLT patterns. The second simulation is a method that uses an Advanced Design System 2020 (ADS). In this simulation, pogo pin probe tips are connected to an ideal SOLT termination. The performance difference between these two simulations leads to an estimation of the adjusted calibration set variable. For accurate result comparison, the structure from the pogo pin to the pad of the calibration standard is the same in both simulations, only the SOLT pattern part is different. In two simulations, the S11 results show the difference of delay and impedance in short, open, and load patterns, and the S21 simulation result shows a delay characteristic of the through pattern. Therefore, the difference in results between these two simulations can adjust the calibration set to the values of inductance, capacitance, impedance, and delay, respectively. These adjusted values are chosen as the VNA calibration set input. Comparing the first and second simulation results, the actual values derived for the calibration patterns are as follows: The short pattern exhibits an inductance of 29 pH, while the open pattern shows a capacitance of −4.3 fF. The load pattern’s terminal impedance is determined to be 42 Ω due to the combined effects of chip resistance, soldering, and PCB patterning. For the through pattern, a time delay of 3 ps is generated. The calculated results are summarized in Table 1. Figure 4b shows the results of two simulations with HFSS and ADS. These simulations also include a pogo pin probe structure, which adds a delay of the pogo pin length to S11 and S21 in Figure 4b. This simulation can also check the parasitic components of the pogo pin itself. The blue line shows the result when the calibration pattern structure is contacted with the pogo pin probe using HFSS, and the red line shows the result when the adjusted value is contacted with the probe using ADS. The simulation results of the two conditions are almost consistent. Since the adjusted value simulation does not include a parasitic component, the result that the two results are almost identical shows that the parasitic component generated by the pogo pin probe itself is very small. In conventional RF probes, and calibration patterns as well, factors such as pad extension, actual resistance impedance, and parasitic capacitance contribute to variations in values, which result in inductance, capacitance, and delay. Typically, even in commercialized products, the short pattern results in a positive series inductance. In contrast, the open pattern’s capacitance varies depending on the structure, leading to either positive or negative capacitance values depending on the product.

Figure 4c,d show the calibration pattern measurement results after applying the ideal and adjusted values from Table 1 into the VNA. When using the ideal values, the pogo pin probe measurement of each SOLT pattern does not reflect the actual characteristics of the calibration pattern. Unlike the actual pattern characteristics, short and open patterns have no delay, load has 50 Ω, and through has no delay. Using these results can reduce the accuracy of Device Under Test (DUT) measurements. In contrast, when using the adjusted values, the through delay becomes clearly visible, and the short and open patterns exhibit changes that align with their actual characteristics. The load pattern measures around 45 Ω, which is closer to the true value. Based on these results, Section 5 presents verification measurements using the results from Figure 4d.

## 5. Pogo Pin Probe Validation with Commercial Integrated Chip

To evaluate the circuit measurement performance of the probe, a measurement setup is configured, as shown in Figure 5a, using a commercial circuit with verified performance. The circuit in Figure 5b is a Gallium arsenide (GaAs) amplifier circuit chip (HMC-ALH244-SX) operating with a gain of 12–13 dB in the 24–40 GHz band. The RF circuit with a 200 μm pitch G-S-G pad is selected to allow simultaneous measurement with both a commercial RF probe and the pogo pin probe. The comparative RF probe (ACP40-A-GSG-150) is a 40 GHz probe manufactured by Cascade Inc., featuring a G-S-G array with a 150 μm pitch. It offers low insertion loss, approximately 0.33 dB up to 45 GHz.

The pogo pin probe is calibrated by a calibration set, in Table 1, with a manufactured calibration standard, and the cascade probe is calibrated using the commercial 101–190 calibration standard manufactured by Cascade. After calibration, the measurement results of the commercial amplifier, shown in Figure 5c,d, are obtained. Figure 5c shows the S21 performance of the amplifier chip measured by the commercial RF probe and pogo pin probe. The solid line represents the pogo pin probe card results, and the dashed line corresponds to the commercial probe measurements. Overall, the S21 magnitude results in Figure 5c show good agreement between the two methods. In addition, the pogo pin probe is repeatedly measured under the same conditions. These results are represented by the red line and the blue line of Figure 5c, and similar results are provided even when repeated measurements are performed. However, when measured with the pogo pin probe, additional ripples appear, and the amplifier’s performance in the high-frequency range is slightly lower. This ripple is presumed to be due to the self-reflection characteristics of the pogo pin probe. As discussed in Section 3, internal transition structures within the pogo pin probe induce reflections, leading to periodic ripples in the measured results. Furthermore, internal probe reflections can impact the magnitude calibration accuracy. This issue is identified as a fabricating-related problem, as described in Section 3, and can be mitigated through a more refined fabrication process.

The measurement results confirm that up to 40 GHz, the trends observed with conventional RF probes and pogo pin probes are similar. However, above 40 GHz, the pogo pin probe exhibits higher insertion loss, leading to reduced calibration accuracy. As a result, it reduces the reliability of the measurement. As shown in Figure 4d, minor errors in through and short calibration appear in the high-frequency range (above 40 GHz). While the overall trend remains consistent, differences in magnitude occur. The primary cause of insertion loss within the pogo pin probe is the PCB converter, which has the longest signal path. To reduce insertion loss, lower-loss PCB material is required. Achieving low insertion loss performance up to 50 GHz would require a substrate like 5 mil Duroid 5880. However, this substrate is too thin to withstand the pogo pin spring pressure. This issue has been validated through the fabrication with 5 mil Duroid 5880 (Rogers Corp., Chandler, AZ, USA) and measurement. To address this issue, an RF-35 substrate is used instead, despite its higher material loss, as it can withstand the required pressure. In contrast, the S21 phase measurements in Figure 5c match well, confirming accurate phase calibration. Figure 5d shows the S11 and S22 results, which also follow a similar trend.

While some of the performance and calibration accuracy require improvement, the pogo pin probe effectively measures general RF circuits up to 50 GHz. Additionally, it offers an advantage over micro-sized needle tip probes by accommodating slopes or steps in the circuit, where micro-sized needle probes would fail to function. The pogo pin probe successfully maintains measurement accuracy even in such non-planar environments.

## 6. Conclusions

The pogo pin probe proposed in this paper achieves a performance comparable to conventional RF probes while also enabling circuit measurements on angled or stepped surfaces. This capability makes it advantageous for probing complex systems where internal components are mounted with soldering, potentially causing steps in the structure. Compared to traditional RF probes, the pogo pin probe can accommodate a wider range of system structures in the millimeter wave band, reducing design constraints and supporting the development of diverse RF systems. This study is also an attempt to replace conventional RF probes, including weak and expensive tips, with probes including highly durable pogo pin tips. Recently, various multi-port RF circuit measurement systems have been studied in millimeter wave bands using VNA [25], in which a commercial RF probe is commonly used for RF signal input and output. The proposed pogo pin probe can be used for commercial multiport IC measurements with VNA where conventional RF probes are used. Unlike conventional pogo pin probe cards designed for repetitive measurements in automated test systems, the proposed probe is adaptable to universal measurement environments by satisfying three key conditions. Table 2 summarizes the structural characteristics and performance of various pogo pin probe designs. To ensure compatibility with commercial IC measurements, the pin alignment must be linear, and the pitch size must not exceed 0.25 mm. Additionally, the probe tip must be angled rather than perpendicular (90°) to allow microscope inspection. The proposed probe meets all of these requirements. Future research will focus on enhancing the prototype to extend its operating frequency range and improve measurement precision. Additionally, further studies will explore the feasibility of replacing conventional RF probes with pogo pin probes featuring narrower pitch sizes.

## Figures and Tables

**Figure 1 sensors-25-01677-f001:**
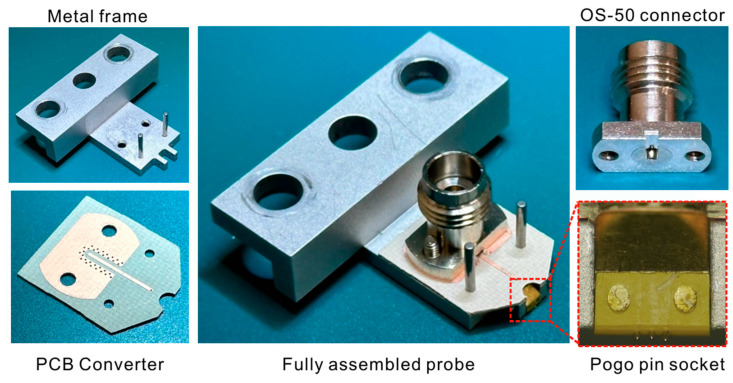
Pogo pin probe with separate photographs of its main components.

**Figure 2 sensors-25-01677-f002:**
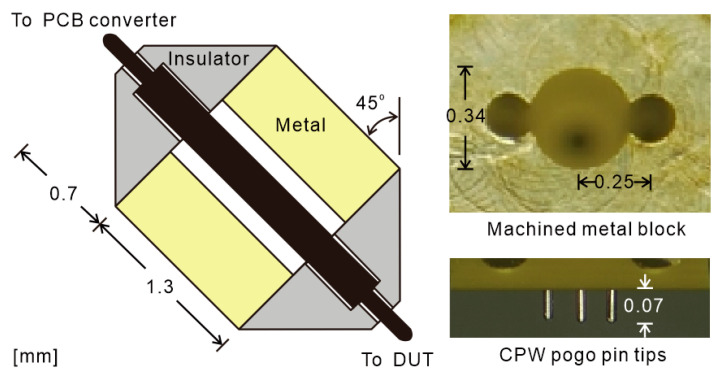
Custom socket design for three pogo pins forming CPW lines with 45° tilt.

**Figure 3 sensors-25-01677-f003:**
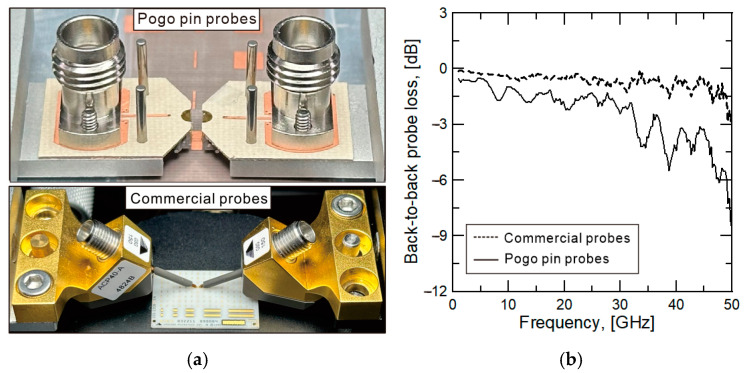
(**a**) Photographs showing loss measurements for back-to-back probes, and (**b**) insertion losses for pogo pin and commercial probes.

**Figure 4 sensors-25-01677-f004:**
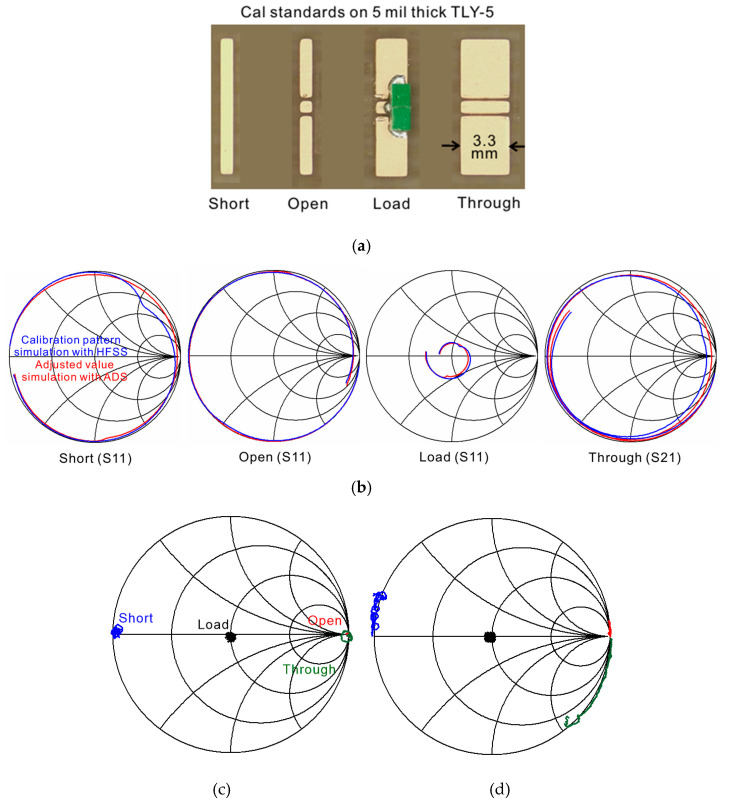
(**a**) Custom calibration standards with 250-micron pitch, (**b**) contacting simulation results of calibration pattern (first simulation) and of adjusted value (second simulation), and calibration results (**c**) with ideal value and (**d**) with adjusted value.

**Figure 5 sensors-25-01677-f005:**
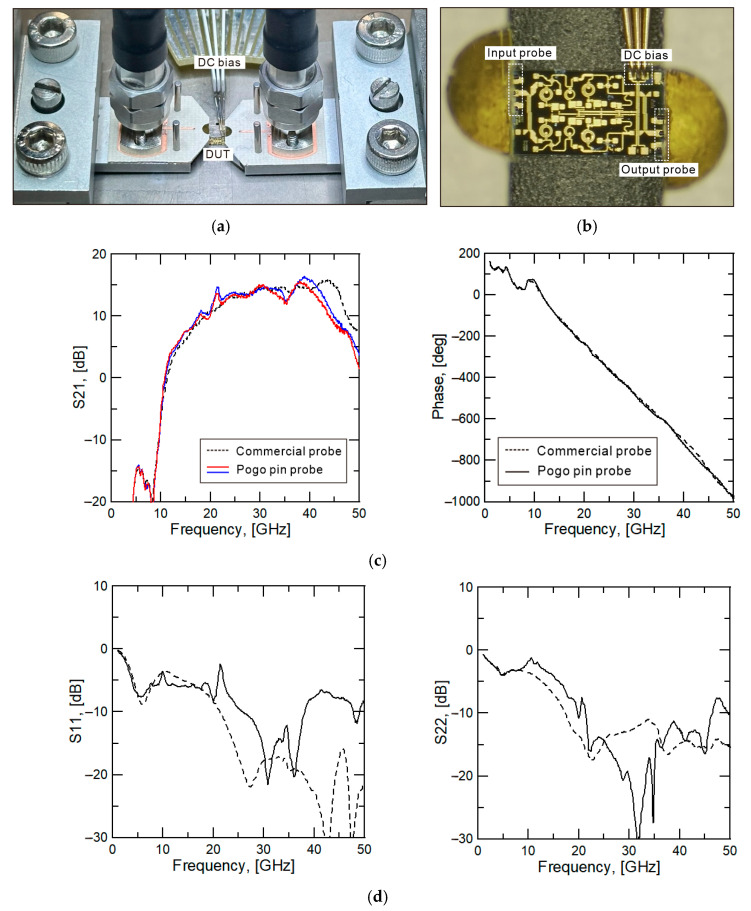
(**a**) Photograph of measurement setup for verifying pogo pin probe performance, (**b**) photograph of commercialized amplifier chip (HMC-ALH244-SX) [24], (**c**) measured S21 results with typical RF probe and pogo pin probe, and (**d**) measured return loss (S11, S22) results.

**Table 1 sensors-25-01677-t001:** Estimated calibration set variables.

Ideal Value	Adjusted Value
Short	29 pH inductance
Open	−4.3 fF capacitance
Load	42 Ω termination imp.
Through (0 ps)	3 ps delay

**Table 2 sensors-25-01677-t002:** Performances of RF probes using pogo pin tip.

Ref.	Max. Frequency	Port	Tip Tilted Angle	Pitch	Pin Alignment	Insertion Loss
[6]	40 GHz	Single	90	2.54 mm	diagonal	6 dB @28 GHz
[11]	20 GHz	Single	90	0.8 mm	-	3 dB @20 GHz
[15]	20 GHz	Single	90	0.616 mm	linear	12 dB @20 GHz
[16]	25 GHz	Single	90	0.35 mm	diagonal	0.5 dB @25 GHz
[17]	50 GHz	Single	90	0.35 mm	diagonal	3.1 dB @50 GHz
[18]	50 GHz	Multi	90	0.35 mm	diagonal	2.1 dB @50 GHz
[22]	40 GHz	Single	90	0.8 mm	linear	0.5 dB @40 GHz
This work	50 GHz	Single	45	0.25 mm	linear	1.6 dB @45 GHz

## Data Availability

Data are contained within the article.

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
