# Peer review of "Construction of CPW Pogo Pin Probes for RFIC Measurements"

_sensors, 2025, doi:10.3390/s25061677_

Round 1

Reviewer 1 Report

Comments and Suggestions for Authors

The authors should provide more detailed comments, including quantitative comparisons, on the novelty of their work in relation to references 17 and 18. These articles appear quite similar; is the primary difference simply a matter of angle and pitch?

Could the authors elaborate on the target ICs and the measurement equipment that require this type of probing? It seems that such probes could be highly useful for modern highly integrated multiport circuits characterized by a VNA, especially since multiport VNAs are now being expanded in terms of functionality (e.g. broadband signal analysis). See, for example: C. Schulze et al., "A VNA-Based Wideband Measurement System for Large-Signal Characterization of Multiport Circuits," IEEE Transactions on Microwave Theory and Techniques, vol. 72, no. 1, pp. 638-647, Jan. 2024. Please discuss this aspect.

In the first sentence, the authors mention "a MEMS tip." What exactly do they mean? Most RF probes do not incorporate MEMS. Is this feature specific to nonplanar circuits? Could the authors cite some commercial models?

When describing the manufacturing process of the "pogo pin probe," the authors should present simulated frequency performance data (not just insertion loss), such as the probe’s S-parameters, to illustrate its inherent parasitics. For instance, parasitic capacitance is likely a relevant factor here. Additionally, radiated fields could pose an issue at these frequencies.

The soldered resistor does not appear to be a robust, standardized component in terms of durability. Moreover, this type of resistor typically exhibits high variability. Was this considered in the study? Why not use standard integrated resistors instead?

It would be valuable to include a study on connection repeatability and compare it to standard coplanar probes.

Comments on the Quality of English Language

The style could be improved although it doesn't prevent understanding.

Author Response

1) The authors should provide more detailed comments, including quantitative comparisons, on the novelty of their work in relation to references 17 and 18. These articles appear quite similar; is the primary difference simply a matter of angle and pitch?

Thank you for the good comment. The references 17 and 18 we previously submitted are prior studies of this study. Pogo pin probe cards from previous references have the disadvantage of being used for RF system measurements only if certain conditions are met. Since the structure was used in a limited environment, there was no need to think deeply about the pogo pin pitch or angle. Instead, the main goal was to design a pogo pin probe card for use at high frequencies without problems. The studies of 17 and 18 are the results of the studies conducted with this goal.

However, for the study to apply the pogo pin structure to general RFIC measurements, another structural improvement was needed. Typical pogo pin probe cards have limited pitch size selection; they can only be used for IC measurements, including RF pads, which are customized to fit the spacing of pogo pin probes, due to mismatch problems and limitations in fabrication.

In this study, the pitch was designed to be 250-micron narrower than previous pogo pin probe card for universal RFIC measurements, and the modified metal socket design was proposed, and insulator structure was improved to solve the problem of narrowing pitch. In addition, the angle of pogo pins was not a consideration because conventional probe cards aim for customized IC measurements using automated systems rather than universal RFIC measurements. However, unlike customized ICs with clear ball maps, universal laboratory and IC test conditions require angle conditions to be considered to avoid misalignment of probe tips. In addition, the durability aspect needs to be considered, so we chose the angle of 45 degrees.

2) Could the authors elaborate on the target ICs and the measurement equipment that require this type of probing? It seems that such probes could be highly useful for modern highly integrated multiport circuits characterized by a VNA, especially since multiport VNAs are now being expanded in terms of functionality (e.g. broadband signal analysis). See, for example: C. Schulze et al., "A VNA-Based Wideband Measurement System for Large-Signal Characterization of Multiport Circuits," IEEE Transactions on Microwave Theory and Techniques, vol. 72, no. 1, pp. 638-647, Jan. 2024. Please discuss this aspect.

This study is an attempt to replace conventional RF probes including weak and expensive tips with probes including highly durable pogo pin tips. The proposed pogo pin probe can be used for general commercial IC measurements where conventional RF probes were used. In this study, we eliminated shortcomings of typical pogo pin probe cards and designed a pogo pin probe to allow pogo pins to be used for general RFIC measurements. For example, the designed pogo pin probe is aimed to replace the RF probe in the measurement system of the reference you provided. Thank you for providing a good reference, and we have added the above contents to the conclusion section.

3) In the first sentence, the authors mention "a MEMS tip." What exactly do they mean? Most RF probes do not incorporate MEMS. Is this feature specific to nonplanar circuits? Could the authors cite some commercial models?

Thank you so much for pointing out the inappropriate expression. Our intention was to explain that the typical RF probe tip is a needle-like structure with a micro-sized structure. But we found out from your review that this word was the wrong choice. Thus, we replaced this expression with ‘micro-sized needle’. Thank you for correcting the wrong expression.

4) When describing the manufacturing process of the "pogo pin probe," the authors should present simulated frequency performance data (not just insertion loss), such as the probe’s S-parameters, to illustrate its inherent parasitics. For instance, parasitic capacitance is likely a relevant factor here. Additionally, radiated fields could pose an issue at these frequencies.

The simulation results you mentioned were added to Figure 4c. The added figure 4c was the result of S11 or S21 when the pogo pin probe was in contact with each calibration standard. (S11 : short, open, load S21 : through) The simulation also includes a pogo pin probe structure, which also shows the performance of the pogo pin probe. The HFSS simulation results show that the performance deformation due to the parasitic component of the probe itself is not significant. It also shows the reason for the selection of the adjusted calibration set variable in Figure 4b compared to the ADS results. This content has also been added to the manuscript along with the figure.

5) The soldered resistor does not appear to be a robust, standardized component in terms of durability. Moreover, this type of resistor typically exhibits high variability. Was this considered in the study? Why not use standard integrated resistors instead?

The soldering resistor we used may have a problem you pointed out. Knowing these problems, we checked the soldering resistance more than 10 samples before using our study. The variation of the resistance used was not large at the range of 50~50.3 W, and the sample closest to 50 W was used for verification. These characteristics of the soldering resistors were confirmed before this study. We also used these soldering resistors to highlight that pogo pin probe structure can be used in structures with height like nonplanar soldering resistors. In future studies, pogo pin gaps between signal and ground are further narrowed to replace typical RF probes. We plan to utilize standard integrated resistance when gaps are narrower. The above sentence has also been added to the manuscript. Thank you for your good opinion.

6) It would be valuable to include a study on connection repeatability and compare it to standard coplanar probes.

We added to Figure 5c the results of repeated measurements with pogo pin probe in the same measurement environment. In Figure 5c , we also compared the S21 results of the commercial standard RF probe commonly used and pogo pin probe. The two results were similar. We have added content to the manuscript as well.

Reviewer 2 Report

Comments and Suggestions for Authors

The manuscript presents an interesting work in the study on pogo pin probe for DC-50GHz application.  It worths for publication but improvement is necessary in the following respects:

  1. In Figure 1, the picture Pogo-pin socket is not clear, it should be replaced by a clearer one.
  2. More marks on Figure 2 will make the description easier. CPW is a key word in this paper, it should be marked in this figure. Especially in the right part of the figure. there are 3 holes with different diameters, it had better appear in this picture.
  3. The title of 3th part of the paper is Pogo pin Probe Fabrications, but it seems not carefully wrote in this part. It more or less describes a measurement result but the fabrication.
Comments on the Quality of English Language

None

Author Response

The manuscript presents an interesting work in the study on pogo pin probe for DC-50GHz application.  It worths for publication but improvement is necessary in the following respects:

1) In Figure 1, the picture Pogo-pin socket is not clear, it should be replaced by a clearer one.

We replaced Figure 1 with a pogo pin socket photograph that looks more clearly.

2) More marks on Figure 2 will make the description easier. CPW is a key word in this paper, it should be marked in this figure. Especially in the right part of the figure. there are 3 holes with different diameters, it had better appear in this picture.

We supplemented the dimensional details by adding a scale bar to the photographs on the right side of Figure 2. Thank you for your detailed comments.

3) The title of 3th part of the paper is Pogo pin Probe Fabrications, but it seems not carefully wrote in this part. It more or less describes a measurement result but the fabrication.

Thank you so much for the detailed point. As you pointed out, section 3 contained a lot of measurement of insertion loss of pogo pin probe. Contents related to fabrication were included in section 1 along with the pogo pin probe design. There was also a distribution of content between sections during the process of manuscript editing, but we accidentally left the title as it was. We modified the title of section 3.

Reviewer 3 Report

Comments and Suggestions for Authors

The author introduces a house made CPW Pogo Pin Probe, claiming it covers a frequency range from DC to 50 GHz. In principle, I agree with this assertion; however, there are notable discrepancies between the manuscript’s writing, actual measurements, and the intended design objectives.

1. The manuscript contains numerous undefined abbreviations. While it is intended for professionals, I strongly recommend that full spellings be provided when abbreviations first appear. For example, terms such as Radio Frequency (RF), MEMS, IC, CST, GSG, and CPW should be fully spelled out initially.

2. The manuscript lacks citations for previous works. For instance, in Line 51, the author does not reference prior studies, which is inappropriate. This issue recurs in Line 70, where no citation is provided for the general calibration standard. Even if the author assumes readers are familiar with these concepts, a detailed description or proper citations are necessary to ensure a solid foundation for the subsequent calibration discussion.

3. The manuscript demonstrates the author’s innovative efforts, which are commendable. However, Figure 2 lacks specific dimensional details, which should be included.  The two images on the right are missing scale bars, which should be added for clarity.

4. There are missing figures. In Line 239, the manuscript references two simulation results (S11, S21), but they are not present. Please include these missing figures to ensure completeness.

Author Response

The author introduces a house made CPW Pogo Pin Probe, claiming it covers a frequency range from DC to 50 GHz. In principle, I agree with this assertion; however, there are notable discrepancies between the manuscript’s writing, actual measurements, and the intended design objectives.

1) The manuscript contains numerous undefined abbreviations. While it is intended for professionals, I strongly recommend that full spellings be provided when abbreviations first appear. For example, terms such as Radio Frequency (RF), MEMS, IC, CST, GSG, and CPW should be fully spelled out initially.

As you pointed out, the first of the abbreviated word is replaced with a full spelled word.

2) The manuscript lacks citations for previous works. For instance, in Line 51, the author does not reference prior studies, which is inappropriate. This issue recurs in Line 70, where no citation is provided for the general calibration standard. Even if the author assumes readers are familiar with these concepts, a detailed description or proper citations are necessary to ensure a solid foundation for the subsequent calibration discussion.

The detailed description of the sentence in line 51 is described in the paragraph containing this sentence, including several references. However, as you pointed out, this sentence could not contain a detailed description, so we modified this sentence. We have added a reference to reinforce the sentence on line 71, and we have also revised the sentence. Thank you for the good point.

3) The manuscript demonstrates the author’s innovative efforts, which are commendable. However, Figure 2 lacks specific dimensional details, which should be included.  The two images on the right are missing scale bars, which should be added for clarity.

We supplemented the dimensional details by adding a scale bar to the photographs on the right side of Figure 2. Thank you for your detailed comments.

4) There are missing figures. In Line 239, the manuscript references two simulation results (S11, S21), but they are not present. Please include these missing figures to ensure completeness.

The simulation results you mentioned were added to Figure 4c. The added figure 4c was the result of S11 or S21 when the pogo pin probe was in contact with each calibration standard. (S11 : short, open, load S21 : through) The simulation also includes a pogo pin probe structure, which also shows the performance of the pogo pin probe. The HFSS simulation results show that the performance deformation due to the parasitic component of the probe itself is not significant. It also shows the reason for the selection of the adjusted calibration set variable in Figure 4b compared to the ADS results. This content has also been added to the manuscript along with the figure.

Round 2

Reviewer 1 Report

Comments and Suggestions for Authors

The authors have responded to the comments, and overall, the article is readable and suitable for publication.

However, the characterization and testing of the pogo-pin probe provided in this article are still too superficial to convincingly demonstrate that it can replace a regular RF probe. I would remove the mention to 50 GHz in the title, which might be misleading. The probe at high frequency is very lossy and would never be used in practice. Also, some key aspects are missing, such as a comprehensive evaluation of the uncertainties associated with this type of probe in typical measurements (e.g., S-parameters) and an accurate performance comparison with regular RF probes. In this regard, the article may have a limited impact.

As a side note, the authors responded that they considered the reference provided, but they forgot to add it to the final manuscript.

Author Response

The authors have responded to the comments, and overall, the article is readable and suitable for publication.

1) However, the characterization and testing of the pogo-pin probe provided in this article are still too superficial to convincingly demonstrate that it can replace a regular RF probe. I would remove the mention to 50 GHz in the title, which might be misleading. The probe at high frequency is very lossy and would never be used in practice. Also, some key aspects are missing, such as a comprehensive evaluation of the uncertainties associated with this type of probe in typical measurements (e.g., S-parameters) and an accurate performance comparison with regular RF probes. In this regard, the article may have a limited impact.

Thank you for your good points. As you pointed out, there was a problem that the measurement accuracy of the pogo pin probe was poor due to high insertion loss at 50 GHz. We modified the '50 GHz' in the title.

We performed S-parameter measurements of the same amplifier in Figure 5 with conventional RF probes and pogo pin probes to show that the two results are similar. As you pointed out, this evaluation alone may be insufficient to show that the pogo pin probe can completely replace the RF probe. However, we tried to show that this measurement allows the proposed pogo pin probe to show its replaceability by showing results similar to those of general RF probes in small signal measurement conditions.

Due to the hard-to-avoid material loss, there is a limit to the pogo pin probes, making it difficult to completely replace the general measurement conditions (e.g., RF power measurement) as you pointed out. We first tried to confirm in this study that pogo pin probes can operate in the same measurement environment through the most basic S-parameter measurements, and we reached this goal. Your good comment became a good guide for our future research. We planned a more improved study in various measurement conditions including large signal condition for replacement RF probe rather than just in a small signal environment. Thank you for the good comments.

2) As a side note, the authors responded that they considered the reference provided, but they forgot to add it to the final manuscript.

Thank you for your detailed review and correcting the mistake. We have added the reference you provided to our manuscript.